# Antimalarial Peptide and Polyketide Natural Products from the Fijian Marine Cyanobacterium *Moorea producens*

**DOI:** 10.3390/md18030167

**Published:** 2020-03-18

**Authors:** Anne Marie Sweeney-Jones, Kerstin Gagaring, Jenya Antonova-Koch, Hongyi Zhou, Nazia Mojib, Katy Soapi, Jeffrey Skolnick, Case W. McNamara, Julia Kubanek

**Affiliations:** 1School of Chemistry and Biochemistry, Georgia Institute of Technology, Atlanta, GA 30332, USA; amsj3@gatech.edu; 2Calibr, a division of The Scripps Research Institute, La Jolla, CA 92037, USA; 3School of Biological Sciences, Georgia Institute of Technology, Atlanta, GA 30332, USA; 4Institute of Applied Sciences, University of the South Pacific, Suva, Fiji; 5Parker H. Petit Institute for Bioengineering and Bioscience, Georgia Institute of Technology, Atlanta, GA 30332, USA; 6Center for Microbial Dynamics and Infection, Georgia Institute of Technology, Atlanta, GA 30332, USA

**Keywords:** marine, cyanobacteria, malaria, mechanism of action, natural product

## Abstract

A new cyclic peptide, kakeromamide B (**1**), and previously described cytotoxic cyanobacterial natural products ulongamide A (**2**), lyngbyabellin A (**3**), 18*E*-lyngbyaloside C (**4**), and lyngbyaloside (**5**) were identified from an antimalarial extract of the Fijian marine cyanobacterium *Moorea producens*. Compound **1** exhibited moderate activity against *Plasmodium falciparum* blood-stages with an EC_50_ value of 8.9 µM whereas **2** and **3** were more potent with EC_50_ values of 0.99 µM and 1.5 nM, respectively. Compounds **1**, **4**, and **5** displayed moderate liver-stage antimalarial activity against *P. berghei* liver schizonts with EC_50_ values of 11, 7.1, and 4.5 µM, respectively. The threading-based computational method FINDSITE^comb2.0^ predicted the binding of **1** and **2** to potentially druggable proteins of *Plasmodium*
*falciparum*, prompting formulation of hypotheses about possible mechanisms of action. Kakeromamide B (**1**) was predicted to bind to several *Plasmodium* actin-like proteins and a sortilin protein suggesting possible interference with parasite invasion of host cells. When **1** was tested in a mammalian actin polymerization assay, it stimulated actin polymerization in a dose-dependent manner, suggesting that **1** does, in fact, interact with actin.

## 1. Introduction

Current malaria drugs are inadequate for future control of the disease due to the evolution of resistance by the malaria parasite, necessitating the discovery of medicines with novel mechanisms of action (MOA) [1,2]. Additionally, pharmaceutical options remain limited for the liver-stage of malaria infection. Natural products such as artemisinin discovered by Nobel-prize winner Tu Youyou have served as valuable sources of drugs to treat malaria [3,4]. Promising nanomolar antimalarial activity has been reported for a wide range of marine-derived natural products including the polycyclic aromatic polyketide trioxacarcin A from a marine *Streptomyces* sp. [5], the polyketide-derived cyanobacterial polyhydroxy macrolide bastimolide A [6], and the β-carboline alkaloid manzamine A produced by a bacterial sponge symbiont [7]. Two sponge-derived antimalarial terpene isonitriles, diisocyanoadociane and axisonitrile-3, were predicted based on molecular dynamics simulations to interfere with heme detoxification, the same MOA as chloroquine [8]. Evolution of an efflux pump that prevents drug accumulation in the parasite’s food vacuole has resulted in resistance of *Plasmodium* to chloroquine [9], illustrating the need for discovery of natural products with unique molecular targets to support future disease management.

Marine cyanobacteria produce a wide variety of secondary metabolites, including peptides, polyketides, alkaloids, lipids, glycosidic macrolides, and terpenes [10], with diverse bioactivities such as antibacterial, antifungal, anticancer, immunosuppressive, and anti-inflammatory [11]. Although the chemical space of this phylum has been extensively studied with over 800 natural products reported in the MarinLit™ database, treatments for parasitic infections have been relatively understudied, leading to opportunities to leverage the diverse chemistry of marine cyanobacteria for pharmacological exploration. The pursuit of antiparasitic natural products has led to promising discoveries, such as the antimalarial cyanobacterial peptides carmabin A [12], gallinamide A [13], and venturamide A [14]. Exploration of new chemical entities and novel bioactivities of known compounds constitute promising routes for inspiring the development of new drug molecules.

An extract of the marine cyanobacterium *Moorea producens* collected off the Northern Lau Islands of Fiji exhibited strong potency against the human malarial parasite *Plasmodium falciparum* and low toxicity to human liver cells. The goal of the present study was to characterize the natural products responsible for the observed antimalarial effects and to explore the intersection of blood-stage and liver-stage activities. These molecules were evaluated for possible protein binding targets using computational predictions generated by the program FINDSITE^comb2.0^ [15,16], a threading-based, virtual ligand-screening method that takes advantage of binding site conservation for evolutionarily distant proteins, and one of the putative molecular targets was confirmed via in vitro testing.

## 2. Results and Discussion

### 2.1. Molecular Structures of Natural Products from Moorea producens 

A Fijian collection of cyanobacteria identified by 16S rRNA sequencing as *Moorea producens* (Appendix A) was freeze-dried, extracted, and the resulting extract was subjected to vacuum liquid chromatography. Bioassay-guided fractionation using both blood-stage *Plasmodium falciparum* and liver-stage *Plasmodium berghei* pointed to four adjacent fractions that demonstrated potent blood and liver-stage antimalarial activity. Subsequent purification of compounds from these fractions using solid-phase extraction (SPE) and high-performance liquid chromatography (HPLC) led to the isolation of **1**–**5** (Figure 1).

High-resolution electrospray ionization mass spectrometry (HRESIMS) was used to assign the molecular formula of **1** as C_42_H_58_N_6_O_7_S with 17 degrees of unsaturation from the pseudomolecular ions [M + H]^+^ at *m/z* 791.4150 and [M + Na]^+^ at *m/z* 813.3958. This matched the molecular formula of the known natural product kakeromamide A (**6**) (Figure 2A) [21], but differences in nuclear magnetic resonance (NMR) chemical shifts indicated unique bond connectivities. The ^1^H NMR spectral data (Table 1) suggested a peptide based on the presence of three amide protons (δ_H_ 6.32, 8.61, and 8.71), two *N*-methyl singlets (δ_H_ 2.87 and 3.02), and four *α*-protons (δ_H_ 4.30, 5.23, 5.45, and 5.63), nearly identical to **6**. In **6,** two methyl doublets at δ_H_ 0.78 and 0.94 corresponded to valine, missing in **1**, while the ^1^H NMR spectrum of **1** had a methyl doublet at δ_H_ 1.40 absent in **6**. 2D NMR spectroscopic analysis using COSY, HSQC, and HMBC experiments identified these differing subunits as an alanine-derived thiazole carboxylic acid and a 3-amino-2-methyloctanoic acid (Amoa), distinct from the valine-derived thiazole carboxylic acid and 3-amino-2-methylhexanoic acid (Amha) in **6**. 2D NMR spectroscopy also confirmed the presence of valine and two *N,O*-dimethyl-tyrosines. The linear peptide sequence was established based on HMBC correlations between α-protons, *N*-methyls, or NHs and carbonyl carbons (Figure 2B; Table 1: H-2/C-1, H-39/C-1 and C-7, H-40/C-6 and C-16, NH-28/C-15, H-28/C-27, H-26/C-25 and C-27, NH-32/C-24, and NH-2/C-30). The relative and absolute configurational assignments in **1** are proposed based on the similarity of chemical shifts and *J* couplings to **6**, which was subjected to a modified Marfey’s method to establish absolute configuration [21]. In the current work, acid hydrolysis towards Marfey’s analysis was attempted, but yields were inadequate for confirming absolute configuration, given that most isolated **1** was needed for pharmacological assessment. 

Compounds **2**–**5** were identified by comparison of HRESIMS, ^1^H NMR, and COSY spectral data to the literature [17,18,19,20] (data provided in Appendix A). Total syntheses have been reported for **2**–**4**, confirming the structures of **2 [22]** and **3 [23]** while leading to the structural revision of three stereogenic centers (C-10, 11, and 13) of **4 [24]**. Two natural products that appeared to belong to the lyngbyabellin class of natural products, referred to as lyngbyabellin-like 1 (**LYN1**) and 2 (**LYN2**), were isolated in very low yields (22 and 28 µg, respectively) which hindered full structure elucidation. Similarities to known lyngbyabellins included the presence of two chlorines as determined from MS isotopic patterns, two downfield proton singlets (**LYN1**: δ_H_ 8.12, 8.16; **LYN2:** δ_H_ 8.11, 8.27) suggestive of thiazole moieties, and methyl singlets with chemical shifts matching the methyl adjacent to the dichloromethylene in **3** (**LYN1**: δ_H_ 2.04; **LYN2:** δ_H_ 2.07). Since these natural products exhibited promising blood-stage antimalarial activity (Table 2), HRESIMS and ^1^H NMR spectral data are included in the Appendix A section even though complete structures were not determined.

### 2.2. Biological Activities of Cyanobacterial Natural Products

The antimalarial activities of **1**–**5** were evaluated against asexual blood-stage *Plasmodium falciparum* and liver-stage *P. berghei* assays while cytotoxicity was measured with HEK293T and HepG2 human cell lines (Table 2). Lyngbyabellin A (**3**) exhibited remarkable activity against blood-stage *P. falciparum* with a half-maximal effective concentration (EC_50_) value of 1.5 nM. Additionally, **3** exhibited a high selectivity index of 13,000 and 2,200 for *Plasmodium* versus normal kidney epithelial cells (HEK293T) and human liver carcinoma cells (HepG2), respectively. Lynbyabellin A (**3**) was as potent as atovaquone (Table 2), indicating that it could be a promising antimalarial drug candidate, especially since a route to production via total synthesis exists [23]. Lyngbyabellins **LYN1** and **LYN2** also exhibited promising activity against blood-stage *P. falciparum* with EC_50_ values of 0.073 and 1.10 μM, respectively. To date, 18 lyngbyabellins have been reported, all structurally related to dolabellin [25] with two disubstituted thiazole rings and a dichloro substituted aliphatic chain [18,19,26,27,28,29,30,31]. Lyngbyabellins possess varying degrees of cytotoxicity against cancer cell lines but had not previously been evaluated for antimalarial properties. Due to the promising antimalarial activities reported in the current study (Table 2), future experiments could evaluate antimalarial mechanisms of action and structure-activity relationships of lyngbyabellins.

Although less potent, cyclic depsipeptide **2** still exhibited promising blood-stage antimalarial activity with an EC_50_ of 0.99 µM, whereas cyclic peptide **1** was only moderately active and **4** and **5** were inactive (Table 2). Evaluation of liver-stage antimalarial activity indicated that **1**, **4**, and **5** are moderately inhibitory against *P. berghei* liver schizont development (sourced from UGA) with EC_50_ values of 11, 7.1, and 4.5 μM, respectively. These same natural products were less active (>12.5 µM) against *P. berghei* sourced from UCSD. None of these natural products exhibited cytotoxicity against HEK293T or HepG2 human cell lines. Characteristics of the ideal drug candidate for treating malaria include safety, especially for vulnerable populations such as children under the age of five, and targeting more than one *Plasmodium* life stage [32]. Although **1** exhibited only moderate antimalarial activity, its ability to inhibit both the blood and liver life stages of *Plasmodium* coupled with its low cytotoxicity towards human cell lines make it a promising lead for drug discovery.

### 2.3. Putative Molecular Targets of Cyanobacterial Natural Products

Modeling of protein-ligand interactions with the FINDSITE^comb2.0^ [16] prediction algorithm suggested that numerous *Plasmodium* proteins are likely to interact directly with these cyanobacterial natural products. Several malarial protein classes were predicted to bind to kakeromamide B (**1**) including actin-like proteins, sortilin, and glutamyl-tRNA(Gln) amidotransferase subunit A. Ranking of protein-ligand interactions by mTC, a composite Tanimoto Coefficient based on fingerprint for ligand similarity measurement, and precision values indicated that actins had the highest probability of binding **1** (Table 3). One of the *Plasmodium* actins predicted to bind with **1**, actin-1, has been found to be involved with host cell invasion [33]. Since the malarial parasite life cycle requires entry into human erythrocytes, inhibiting the ability of *Plasmodium* to invade host cells could be a viable mechanism of action. The malarial protein with the next highest mTC and precision values for binding **1**, sortilin, is important for the formation of apical complex components and creation of merozoites [34]. In parasite strains where the expression of sortilin is suppressed, new merozoites, which are the invasive form of the parasite, cannot be formed. Thus, like actin, inhibition of sortilin could impact *Plasmodium’s* ability to infect host cells suggesting that this protein could also be a feasible target for future exploration. Glutamyl-tRNA(Gln) amidotransferase subunit A is essential for the parasite’s blood-stage [35]. This protein is required for the synthesis of glutamyl-tRNA-gln, a protein synthesis building block needed for parasite propagation [36]. Since the precision value for glutamyl-tRNA(Gln) amidotransferase subunit A was relatively low, this target is less promising. 

Of the proteins predicted to bind to kakeromamide (**1**), actin and sortilin were deemed worthy of further evaluation given their higher predicted binding values. However, no in vitro sortilin assay is available. In contrast, actin polymerization assays are frequently employed to assess agents that affect actin dynamics, albeit using mammalian homologs of actin. Since **1** was predicted to bind to both *Plasmodium* and mammalian actin proteins but only mammalian actin assays are commercially available, an actin polymerization assay that uses pyrene-labeled rabbit muscle actin was employed. Several natural products are known to disrupt actin’s function by either inhibiting or stabilizing polymerization, which could arise from ligand binding [37,38,39,40]. Since apicomplexan actins (including those of *Plasmodium*) and mammalian actins are relatively divergent [41], *Plasmodium* actin might be a useful molecular target. Additionally, since **1** did not exhibit toxicity towards human cell lines (Table 2), the predicted interactions with human actin apparently did not translate to a negative outcome for host cells, at least in vitro. The effect of **1** on rabbit actin polymerization was compared to the known actin polymerization inhibitor, latrunculin A. While latrunculin A suppressed polymerization at concentrations above 1.2 µM, **1** exhibited unique behavior with the highest tested concentration, 25 µM, enhancing actin polymerization whereas the lowest tested concentration, 0.025 µM, moderately suppressed polymerization relative to controls (Appendix A). This suggests that **1** causes a concentration-dependent modulation in mammalian actin function. Cyclic peptides such as phalloidin [37] from the death cap mushroom and jasplakinolide [40] from a marine sponge have been found to induce polymerization and stabilize actin filaments in a dose-dependent manner. Similar activity by **1** could be one possible explanation for the observed increase in actin polymerization with increasing concentrations of **1.** Although **1** appears to impact mammalian actin polymerization, future tests evaluating the impact of **1** on *Plasmodium* actin function would be worthwhile in order to further elucidate whether interactions with actin are relevant to this natural product’s mechanism of action on the malaria parasite.

Ulongamide A (**2**) was predicted to bind to one protein, an aspartyl protease, with mTC and precision values of 0.47 and 0.35, respectively. Aspartyl protease is involved with remodeling the erythrocyte by trafficking parasite proteins into erythrocytes, which ultimately aids the parasite in evading host recognition and response [42]. In fact, targeting aspartyl proteases in HIV has led to the development of clinically available antiretrovirals, which have also shown inhibition towards *Plasmodium* growth [43]. Taken together, aspartyl protease seems like a potentially promising target for **2**.

Lyngbyabellin A (**3**), the most potent antimalarial natural product isolated from *Moorea producens* in the current study, did not deliver any putative molecular targets via FINDSITE^comb2.0^ protein-ligand binding calculations. Even a linear analog of **3**, lyngbyabellin I [27], was not predicted to bind to *Plasmodium* proteins. It is possible that the structural motifs of **3** are inadequately represented among ligands known to bind malarial proteins as sampled from the literature; however, **3** was predicted to bind to several human proteins. Thus, there exist compounds structurally similar enough to **3** in ligand databases to enable prediction of some protein binding events, but apparently not with *Plasmodium* proteins. We are, therefore, unable to propose hypotheses of possible molecular targets of **3**. The promising selectivity of **3** towards *Plasmodium*, the availability of a total synthesis route, and the substantial collection of known lyngbyabellins warrants further study as a possible treatment option for malaria. 

## 3. Materials and Methods 

### 3.1. General Experimental Procedures

^1^H, COSY, HSQC, and HMBC NMR spectra were acquired on either a 16.4 T (700 MHz for ^1^H and 175 MHz for ^13^C) Bruker Advance III HD instrument with a 5mm indirect broadband cryoprobe or an 18.8 T (800 MHz for ^1^H and 200 MHz for ^13^C) Bruker Avance III HD instrument equipped with a 3 mm triple resonance broadband cryoprobe. Spectra were recorded in CD_3_CN or CDCl_3_ and referenced to solvent residual peaks (δ_H_ 1.94 and δ_C_ 118.26 or δ_H_ 7.26 and δ_C_ 77.16, respectively). Spectra were processed and analyzed using MestReNova 12.0.0. Vacuum liquid chromatography (VLC) was performed using SiliCycle’s SiliaFlash F60, 40−63 μm silica gel. Supelco Supelclean ENVI-18 was used for solid-phase extraction (SPE). High-performance liquid chromatography (HPLC) separations were done with a Waters 1525 binary pump and either a Waters 2487 dual wavelength absorbance detector set at 254 nm or a Waters 2996 photodiode array detector, using two different columns: a 4.6 × 250 mm, C18-silica reversed-phase (Grace Alltima, 5 μm particle size) and a 4.6 × 250 mm phenyl-hexyl phase (Phenomenex Luna, 5 μm particle size). High-resolution mass spectrometry (HRMS) data were collected with a ThermoFisher Scientific LTQ Orbitrap XL ETD spectrometer in positive ion mode. The masses of isolated natural products were estimated by quantitative NMR (qNMR) using a capillary filled with benzene-d6 as an internal standard that was placed inside an NMR tube containing a known amount of caffeine and tubes containing the identified compounds dissolved in the same amount of solvent [44].

### 3.2. Biological Material and Species Identification

The dark purple marine cyanobacteria were found carefully organized into curtains covering snapping shrimp holes located on reef slopes offshore of Tuvuca Island in Fiji. The cyanobacteria were collected on September 25, 2007 from two separate locations (S 17°44.43′, W 178°39.155′ and S 17°30.546′, W 178°42.443′) and combined based on similarity in sample appearance and habitat. Voucher specimens identified as collection G-0362 were stored at the University of the South Pacific and Georgia Institute of Technology in aqueous ethanol and formaldehyde. Bulk cyanobacterial sample was stored at the Georgia Institute of Technology in a −80 °C freezer.

The cyanobacterium was identified as *Moorea producens* through morphological and 16S rRNA phylogenetic analyses. Briefly, the ethanol-preserved genomic DNA of cyanobacterial specimen G-0362 was extracted with the DNeasy Blood and Tissue kit (Qiagen) using the manufacturer’s protocol. Polymerase chain reaction (PCR) was utilized to amplify the single 16S rRNA gene fragment from genomic DNA using bacterial 16S rRNA primers, 27f, and 1492r [45]. A 25 μl reaction volume was used for PCR amplification with the following components: 200 μM of each dNTP, 50 ng of purified genomic DNA, 1 μM of each oligonucleotide primer, 1.0 U Taq DNA Polymerase, and 1× Standard PCR reaction buffer (NEB, Ipswich, MA). A GeneAmp PCR system 2700 thermocycler (Applied Biosystems, Foster City, CA) was used to perform the PCR amplification with the following parameters for temperature cycling: 94 °C for 4 min for initial denaturation followed by 40 cycles of amplification (each cycle = denaturation for 50 s at 94 °C, primer annealing for 50 s at 55 °C, and primer extension for 2 min at 72 °C). The incompletely synthesized DNA underwent final extension at 72 °C for 7 min. Agarose gel electrophoresis (1% wt/vol) was used to analyze the PCR product, and an ethidium bromide stain was applied before the gel was visualized with a UV transilluminator. Forward and reverse primers were utilized to sequence the PCR product and the CAP3 Sequence Assembly Program was used to manually edit and assemble sequences [46]. The assembled 16S rRNA sequence (1338 bp) was submitted to GenBank (accession no. MN960022). The blastn program was used to assess the sequence similarity of the assembled contig of cyanobacterial 16S rRNA to other known cyanobacteria from Family Oscillatoriaceae by comparing it with the non-redundant nucleotide database (NCBI) [47]. To determine the closest phylogenetically related species of *Moorea producens*, the 16S rRNA sequence of collection G-0362 was compared with those of known representatives from Family Oscillatoriaceae (Order Oscillatoriales) and *Gloeobacter violaceous* was included as a distant evolutionary taxon. MEGA X [48] was used for phylogenetic analysis using the Maximum Likelihood method based on the Kimura 2-parameter model [49] with 1000 bootstrap iterations.

### 3.3. Extraction and Isolation

The cyanobacterial biomass (43 g) was freeze-dried (to 8.0 g) and exhaustively extracted (four times) using a 2:1 mixture of dichloromethane and methanol. The resulting crude extract (0.80 g) was purified by VLC with 80 mL of silica gel in a glass vacuum funnel (8 cm diameter by 9 cm high) with a glass frit. The fractions were collected using a stepped gradient of hexanes, EtOAc, and MeOH (nine fractions, 100% hexanes, 10% EtOAc/90% hexanes, 20% EtOAc/80% hexanes, 40% EtOAc/60% hexanes, 60% EtOAc/40% hexanes, 80% EtOAc/20% hexanes, 100% EtOAc, 25% MeOH/75% EtOAc, and 100% MeOH). The fractions eluting from 40% EtOAc/60% hexanes (14.1 mg), 60% EtOAc/40% hexanes (7.4 mg), 80% EtOAc/20% hexanes (5.8 mg), and 100% EtOAc (3.7 mg) were further separated using reversed phase SPE (stepwise gradient of decreasing polarity starting with 15% MeOH in H_2_O to 100% MeOH, resulting in five fractions). Fractions 3, (50% MeOH in H_2_O, 2.4 mg), 4 (75% MeOH in H_2_O, 2.0 mg) and 5 (100% MeOH, 4.2 mg) were subjected to C_18_-silica HPLC (50% MeCN/50% H_2_O held for 5 min, gradient from 50% MeCN/50% H_2_O to 100% MeCN from 5 to 40 min, and 100% MeCN from 40 to 60 min at 1 mL/min) to obtain nearly pure **1**–**5**. Fractions containing **1** and **3** underwent one more round of purification with C18-silica HPLC (60% MeCN/40% H_2_O + 0.05% trifluoroacetic acid (TFA) held for 5 min, gradient from 60% to 70% MeCN from 5 to 20 min, and 70% MeCN held from 20 to 25 min at 1 mL/min) to give **1** (*t*_R_ = 17.5 min, 0.18 mg) and **3** (*t*_R_ = 12.2 min, 0.069 mg). The same C_18_-silica HPLC column was also used to purify **2** (isocratic method of 50% MeCN/50% H_2_O + 0.05% TFA held for 25 min at 1 mL/min), resulting in pure **2** (*t*_R_ = 18.5 min, 0.34 mg). A mixture containing **4** and **5** was purified using phenyl-hexyl silica HPLC (gradient from 80% MeOH/H_2_O to 90% MeOH/H_2_O over 20 min at 0.8 mL/min) to give **4** (*t*_R_ = 14.5 min, 0.20 mg) and **5** (*t*_R_ = 15.7 min, 0.083 mg). The reported isolated yields do not include losses during separation or material subjected to bioassays and thus underestimate natural concentrations. 

Kakeromamide B (**1**): ^1^H and ^13^C NMR data, see Table 1; HRMS (ESI) *m/z* [M + H]^+^ calcd for C_42_H_59_N_6_O_7_S 791.4166, found 791.4150; [M + Na]^+^ calcd for C_42_H_58_N_6_O_7_SNa 813.3985, found 813.3958.

### 3.4. Antimalarial and Cytotoxicity Assays

A standard method for evaluating antimalarial activity with asexual blood-stage *P. falciparum* and SYBR Green detection was used to assess the potency of extracts, fractions, and compounds [50]. Screening media was prepared from complete media supplemented with 0.05% Albumax II (but without human serum). Dr. David Fidock of Columbia University provided *P. falciparum* strain Dd2L, which was cultured with fresh O+ erythrocytes (TSRI Normal Blood Donation). Compounds were transferred to assay plates using a Labcyte ECHO acoustic liquid handler. Fresh and parasitized erythrocytes were prepared in screening media before inoculating plates (Multi-Flo; BioTek, VT) with a final hematocrit of 2.5% and a final parasitemia of 0.3%. The assay plates were incubated for 72 h in a chamber at 37 °C with a low oxygen gas formulation with daily gas exchanges. After incubation, a Multi-Flo liquid dispenser transferred SYBR Green lysis buffer to the wells, and the plates were incubated for another 24 h at room temperature to achieve optimal development of fluorescence signal, which was read by an Envision Multimode Reader (PerkinElmer, MA). Atovaquone, ganaplacide, and mefloquine were used as positive controls (EC_50_ = 0.00040, 0.0059, and 0.0029 μM, respectively).

Activity against liver-stage *P. berghei* was quantified for extracts, fractions, and pure compounds using a slightly modified 48 h luminescence-based assay [51]. The salivary gland of infected *Anopheles stephensi* was dissected to harvest *P. berghei*-ANKA-GFP-LucSMCON (Pb-Luc) sporozoites (originally received from Dr. Ana Rodriguez of NYU School of Medicine; last round of testing to evaluate pure **1, 3**–**5** used infected mosquitoes from Elizabeth Winzeler at UCSD and Dennis Kyle at UGA) which were filtered through a 20 μm nylon net filter twice (Steriflip, Millipore, MA), counted with a hemocytometer, and adjusted to achieve a final concentration of 200 sporozoites/1 μL in the assay media (DMEM without Phenol Red (Life Technologies, CA), 5% FBS, and 5 × Pen−Strep and glutamine (Life Technologies, CA)). A Multi-Flo (BioTek, VT) with a 1 μL cassette was used to transfer purified sporozoites, 1 × 10^3^ sporozoites per well (5 μL), to assay plates containing HepG2 cells. After incubation at 37 °C for 48 h, a bioluminescence measurement was taken to assess the liver-stage growth. More specifically, media was removed by inverting the plates, and spinning them at 150 × g for 30 s, and 2 μL per well of BrightGlo (Promega, WI) was added for quantification of Pb-Luc viability. The luminescence was measured immediately after addition of the luminescence reagent using an Envision Multilabel Reader (PerkinElmer, MA).

A previously described method was utilized to evaluate cytotoxicity against HepG2 and HEK293T cells as a measure of general human toxicity [52]. Assay data generated for *Plasmodium* was analyzed with a Genedata Screener (v13.0-Standard). Inhibitors (puromycin for cytotoxicity and atovaquone for liver-stage parasites) were subtracted from neutral controls to normalize data. The Smart Fit function of the Genedata Analyzer generated dose-response curves that were used to calculate the half-maximal effective concentrations (EC_50_). 

### 3.5. Computational Binding Predictions

FINDSITE^comb2.0^ [16] was applied to screen cyanobacterial natural products against the whole genome of *P. falciparum* (isolate 3D7) (PF). FINDSITE^comb2.0^ builds structural models using a threading approach by taking the protein amino acid sequence as input and applying TASSER refinement [53]. Models were built for 4,962 (92.5%) protein sequences out of a total of 5,364 PF sequences downloaded from UniProt (www.uniprot.org). For each target protein, the pockets were identified in their respective models and compared to the ligand-binding pockets located in PDB [54] structures as well as ChEMBL [55] and DrugBank [56] libraries. Pockets that best matched target pockets were selected, and their corresponding binding ligands were employed as template ligands. These template ligands were used for virtual screening as seed ligands for a given molecule with a fingerprint comparison method to evaluate the binding probability of the molecule to the protein targets. The mTC virtual screening score, a measure of the similarity of the given molecule to the template ligands of a protein target in FINDSITE^comb2.0^, was used to rank the targets. The mTC correlates with the likelihood of the protein target being a true target of the molecule, and its value ranges between 0 and 1 with 1 indicating the best fit. An mTC of 0.5 corresponds to the expected precision of ~0.76 based on large scale benchmark statistics [16].

### 3.6. Actin Polymerization Assay

The computationally predicted binding of **1** to actin was experimentally evaluated using a rabbit muscle actin polymerization assay kit from Abcam (kit ab239724) using the manufacturer’s protocol. Mean signal for n = 3 (actin polymerization control, solvent control, and background) or n = 2 (four different concentrations of latrunculin A and kakeromamide B) replicates was acquired for time points taken every 30 s for 1 h on a Synergy H4 plate reader. The results were plotted in GraphPad Prism 8 with an exponential plateau curve.

## Figures and Tables

**Figure 1 marinedrugs-18-00167-f001:**
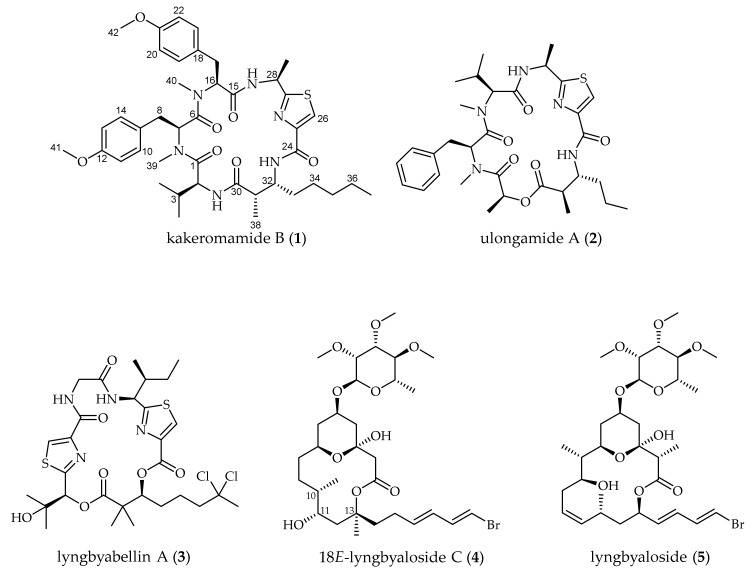
Natural products from the Fijian marine cyanobacterium *Moorea producens*, including the novel cyclic peptide kakeromamide B (**1**) [17,18,19,20].

**Figure 2 marinedrugs-18-00167-f002:**
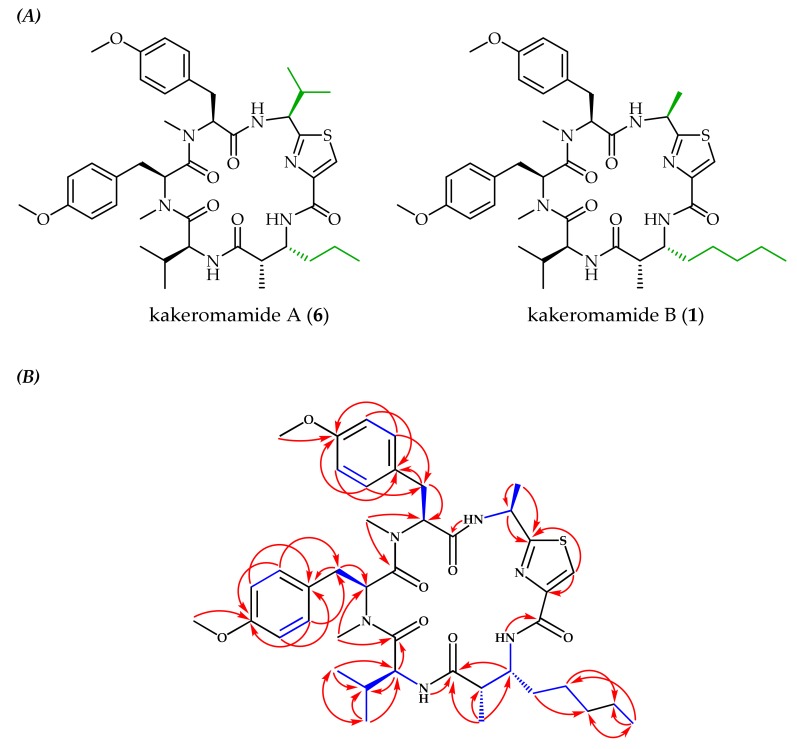
(**A**) Comparison of the known cyclic peptide kakeromamide A (**6**) [21] to the newly identified analog, kakeromamide B (**1**), with distinguishing moieties highlighted green. (**B)** Observed COSY (blue bonds) and HMBC (red arrows) correlations for kakeromamide B (**1**).

**Table 1 marinedrugs-18-00167-t001:** ^1^H (700 MHz) and ^13^C (175 MHz) NMR spectroscopic data of kakeromamide B (**1**) in CD_3_CN.

no.	δ_C_ (mult., *J* in Hz)	δ_H_ (mult., *J* in Hz)	COSY	HMBC
1	176.3			
2	56.9	4.30 t (7.1)	H-3,2-NH	C-1 ^w^,C-3,C-4 ^w^
2-NH		6.32 d (6.6)	H-2	C-30 ^w^
3	30.8	1.80 m	H-2,H-4,H-5	
4	18.6	0.78 d (6.9)	H-3	C-2,C-3,C-5
5	19.0	0.80 d (6.8)	H-3	C-2,C-3,C-4
6	172.1			
7	50.3	5.63 dd (4.9, 10.9)	H-8a,H-8b	
8a	33.2	1.37 m	H-7,H-8b	C-9 ^w^
8b		2.74 dd (11.0, 16.1)	H-7,H-8a	C-7,C-9
9	129.8			
10/14	130.0	6.89 d (8.5)	H-11/13	C-8 ^w^,C-12
11/13	114.7	6.78 d (8.7)	H-10/14	C-9
12	159.1			
15	168.9			
16	62.9	5.23 dd (5.0, 10.0)	H-17a,H-17b	
17a	34.8	2.67 dd (10.1, 14.4)	H-16,H-17b	C-16
17b		2.92 dd (4.9, 14.4)	H-16,H-17a	C-18
18	130.7			
19/23	131.4	7.00 d (8.6)	H-20/22	C-17,C-21
20/22	114.9	6.60 d (8.7)	H-19/23	C-18
21	159.4			
24	169.5			
25	150.2			
26	123.2	8.02 s		C-25 ^w^,C-27
27	171.6			
28	48.1	5.45 p (7.4)	28-NH,H-29	C-27 ^w^
28-NH		8.61 d (7.8)	H-28	C-15 ^w^
29	24.0	1.40 d (6.9)	H-28	C-27,C-28
30	173.6			
31	44.5	2.63 qd (3.7, 6.8)	H-32,H-38	
32	53.1	4.02 m	H-31,32-NH,H-33a,b	C-30 ^w^
32-NH		8.71 d (10.2)	H-32	C-24 ^w^
33a	41.8	1.13 m	H-32,H-33b,H-34a,b	C-35 ^w^
33b		1.72 m	H-32,H-33a,H-34a,b	
34a	30.4	1.29 m	H-33a,b	C-36 ^w^
34b		1.45 m	H-33a,b	
35	33.7	1.35 m		C-37 ^w^
36	24.5	1.35 m	H-37	
37	14.9	0.88 t (6.6)	H-36	C-34,C-35,C-36
38	19.6	1.09 d (6.9)	H-31	C-30,C-31,C-32
39	31.4	3.02 s		C-1,C-7
40	28.7	2.87 s		C-6,C-16
41	56.0	3.74 s		C-12
42	55.7	3.49 s		C-21

^w^ Indicates weak correlation.

**Table 2 marinedrugs-18-00167-t002:** Antimalarial activities and cytotoxicities of natural products from *Moorea producens*.

	Blood-Stage *P. falciparum* ^a^	Liver-Stage *P. berghei* ^a^	HEK293T Cytotoxicity ^b^	HepG2 Cytotoxicity ^b^
**1**	8.9	11 ^d^, >12 ^e^	>23	>23
**2**	0.99	>4.0 ^c^	>4.8	not tested
**3**	0.0015	>10 ^d,e^	19	3.3
**4**	>19	7.1 ^d^, >16 ^e^	>31	17
**5**	>7.9	4.5 ^d^, >6.3 ^e^	>13	>13
lyngbyabellin-like 1 **(LYN1)**	0.073	>3.2 ^e,d^	>6.5	4.5
lyngbyabellin-like 2 **(LYN2)**	1.1	>2.6 ^e,d^	>5.2	>5.2
Atovaquone (+ control)	0.0061	<0.00028 ^c^, 0.0017 ^d^, 0.0037 ^e^	>2.0	not tested

^a^ Half-maximal effective concentration (EC_50_) (μM). ^b^ Half-maximal cytotoxicity concentration (μM). ^c^ Natural product tested against *P. berghei* sporozoites sourced from New York University (NYU) School of Medicine. Later liver-stage *P. berghei* testing used sporozoites from the University of Georgia (UGA) ^d^ and University of California San Diego (UCSD) ^e^.

**Table 3 marinedrugs-18-00167-t003:** Proteins predicted to bind to kakeromamide B (**1**) using the protein-ligand prediction algorithm FINDSITE^comb2.0^ [16]. Higher values for mTC (a Tanimoto Coefficient based fingerprint for similarity measure) and precision correspond to higher probability of the protein being a true target.

Accession Number	Protein Description	mTC	Precision
**Q8ILW9**	Actin-2	0.50	0.77
**Q8ILM5**	Actin-related protein homolog, arp4	0.50	0.77
**Q8IIW6**	Actin-like protein, putative	0.50	0.77
**Q8IIQ4**	Actin-like protein homolog, ALP1	0.50	0.77
**Q8IBH5**	Actin-like protein, putative	0.50	0.77
**Q8I4X0**	Actin-1	0.50	0.77
**Q8I450**	Actin-like protein, putative	0.50	0.77
**Q8I345**	Actin-like protein, putative	0.50	0.77
**Q8I2A2**	Actin-related protein, ARP1	0.50	0.77
**Q8I1V9**	Actin-like protein, putative	0.50	0.77
**Q8IKV8**	Sortilin, putative	0.48	0.45
**Q8I1S6**	Glutamyl-tRNA(Gln) amidotransferase subunit A, putative	0.46	0.26

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
