# Peer review of "Antimalarial Peptide and Polyketide Natural Products from the Fijian Marine Cyanobacterium Moorea producens"

_marinedrugs, 2020, doi:10.3390/md18030167_

Round 1

Reviewer 1 Report

In the paper by Kubaneck et al a new cyclic peptide togheter with known metabolites, isolated from an antimalarial extract of a marine cyanobacterium   are described. Antimalarial activities an cytotoxicity of all compounds were evaluated. Moreover threading-based computational methods were applied  in order to predict the putative molecular targets of active compounds. An in vitro test was performed for the new cyclic peptide. The results prompted the formulation of hypothesis about the possible mechanisms of action.

Structural charachterization, biological tests and computational studies have been properly reported. The paper is of interest for Marine drugs readers and suggests new insights for the discovery of new antimalarial drugs.

Author Response

Thank you for taking the time to review our manuscript and for providing us with positive feedback.

As there were no suggested revisions from reviewer 1, we will not provide a point-by-point response. 

Reviewer 2 Report

This manuscript reports on isolation and identification of natural products, namely, kakeromamide B (1), ulongamide A (2), lyngbyabellin A (3), 18E-lyngbyaloside C (4) and lyngbyaloside (5) from an extract of the Fijian marine cyanobacterium Moorea producens. Two additional compounds, presumably belonging to the lyngbyabellin class: LYN 1 and LYN 2, were also isolated in quantities not sufficient for structure elucidation. The naming of these products should be unified because also alternative abbreviations (LYN1 and LYN2) appear in the manuscript. Kakeromamide B (1) proved to be a novel compound, whereas compounds 2-5 were previously described in the literature. The structure of the first product was unambiguously established by careful analysis of its HRESIMS data and advanced NMR methods (COSY, HSQC, HMBC), whereas structure elucidation of the remaining compounds (2-5) was achieved by comparing their spectroscopic data with these reported in the literature. 

The spectroscopic data are clearly shown in the Tables and all of the 1H and 13C signals are completely assigned. Based on NMR data presented in the Table 1, the structure elucidation of kakeromamide B (1) is convincing. However, two pieces of data from this Table call for the Authors’ attention:

  • Why is the signal of H-3 at 1.80 ppm described as quartet (q)? Splitting caused by six protons from both methyl groups and H-2 (all coupling constants ca. 7 Hz) should result in splitting into an octet! Similar signal of the proton from isopropyl substituent in the spectrum of ulongamide A (2) resonating at 2.33 ppm was described as a multiplet (m).
  • Why are crosspeaks from protons attached to aromatic carbons 19 and 23 listed in the HMBC column as “C-17a,b, C-21”? It should read: C-17, C-21.

The study is strengthened by biological evaluation of the final compounds. Assessment of antimalarial potency of the isolated natural products 1-5, LYN1 and LYN2 indicates promising activities of some of the compounds. Among them, the previously unknown kakeromamide B (1) exhibited a promising activity profile. Consequently, further structure-activity studies involving similar compounds are desirable.

In addition to the judgement, that these studies constitute valid research, the experimental part of the manuscript and Supplementary Materials, providing MS and particularly important NMR spectra of the isolated compounds, as well as their comparison with the respective literature chemical shifts, is also correctly written.

This Reviewer considers the manuscript presented to him as an important and interesting study, potentially interesting for a broad group of scientists and, therefore, strongly recommends the article for publication.

Author Response

We thank reviewer 2 for his careful reading of our manuscript and the helpful feedback that he provided. The attachment contains point-by-point responses to his comments. 
